# Prevalence and associated risk factors of burnout amongst veterinary students in Ghana

**Benjamin Obukowho Emikpe**[1]*, **Derrick Adu Asare**[2], **Abigael Omowumi Emikpe**[3], **Ludwig Albert Nortey Botchway**[1], **Richard Abeiku Bonney**[1]

**1** School of Veterinary Medicine, Kwame Nkrumah University of Science and Technology, Kumasi, Ghana, **2** University of Ghana, Legon, Accra, Ghana, **3** Department of Nursing, Faculty of Allied Health Sciences, Kwame Nkrumah University of Science and Technology, Kumasi, Ghana

* banabis2001@yahoo.com

**Data Availability Statement:** All relevant data are within the paper.

## Abstract

This study was designed as a cross-sectional study to find out the prevalence and associated risk factors of burnout among veterinary students at Kwame Nkrumah University of Science and Technology (KNUST) in Kumasi—Ghana. A total of 74 veterinary students served as the respondents and were given online questionnaires which comprised questions on emotional exhaustion (EE), depersonalisation (DP) and reduced personal accomplishment (RPA). Data obtained were analysed using descriptive statistics, Chi-square test and regression analysis. Results indicated that on average, the EE and RPA were low with mean score of $12.72 \pm 4.46$ and $27.96 \pm 7.94$ whilst DP was high with a mean score of $20.72 \pm 6.5$ among veterinary students. The year of study had a significant effect ($p = 0.000$) on burnout with levels of high RPA (80%) and DP (70%) being more in the preclinical students as compared to the clinical students (RPA = 20%, DP = 30%). There was also a significant effect ($p = 0.028$) of the year of study and type of residence of the students on the high level of DP and RPA. Daily sleep hours of students had Daily sleep hours of the veterinary students were also indicated as a significant associated risk factor of burnout among veterinary students in this study as more less sleep hours (<6hours), causes high levels of DP and RPA. Age, gender and marital status, working status, exercise hours and weekly study hours of students did not count as significant predictive factors ($p > 0.05$) of burnout.

## Introduction

Medicine related programmes worldwide are aimed at training and developing world-class students to be well-equipped with the required skills and attitudes necessary to ensure the health needs of the immediate societies and the world at large are not left in jeopardy [1]. In view of this, medical schools across the world have focused on developing a more robust curriculum and nature of studies for their students so that no stone will be left unturned in the bid to delivering good quality medical education [2] of which veterinary medical schools are

**Funding:** The authors received no specific funding for this work.

**Competing interests:** The authors have declared that no competing interests exist.

no exceptions. The robustness of such medical schools can be very stressful for students and can cause burnout syndrome.

Burnout syndrome is characteristically a combination of negative energies that are manifested in emotional and physical exhaustion, depersonalisation, and reduced personal achievements [3]. The burnout syndrome has been reported more in professionals and workers in the human medical schools [4, 5] with less report on veterinary schools. In the academic setting, burnout has been reported by several researchers to be occurring in students of various universities in different countries [6–8] of which veterinary students are no exception [9–11]. Research has indicated strongly that enrolling in a veterinary school comes with a certain level of exposure to stress. This kind of stress exposure is mostly prolonged due to the lengthy years spent in training as a veterinary student [9]. These stresses could arise from many sources which could be either academic related or non-academic related. Academically, the requirement of universities and the demand on students to maintain very good academic standings to keep them in school, coupled with the heavy workload of the veterinary medicine curriculum causes a relatively high level of stress amongst students [12, 13]. In addition to the demanding curriculum, the amount of time dedicated to learning the vast information on the numerous animal species is another source of academic stress [14]. The non- academic sources of stress include the perceived difficulty in fitting in as well as unclear expectations on the part of students, poorer perceived physical health, financial stress and certain stressful activities that students may be involved in during their course of study among others [15–17]. The excessiveness of these stressors leads to burnout syndrome which negatively affects the academic performance of students.

Though several research works have been reporting burnout among students in the medical and veterinary medical schools, the review of literature indicates a void in literature on the occurrence of burnout amongst veterinary students especially in Africa. There is therefore the need to fill this gap in literature concerning burnout amongst veterinary students. This study therefore is focused on examining the level of burnout amongst veterinary students in Ghana as well as the risk factors that expose veterinary students in Ghana to burnout syndrome. The findings of this study will be key in informing the stakeholders in veterinary education to take pragmatic and proactive measures to deal with burnout syndrome amongst veterinary students in order to enhance their academic well-being.

## Methods

### Research design and study area

This study was designed as a cross-sectional study at the School of Veterinary Medicine (SVM) in Kwame Nkrumah University of Science and Technology (KNUST) in Kumasi—Ghana, one of the two veterinary schools in Ghana. This research was conducted within a month between June and July 2020.

The School of Veterinary Medicine is under the College of Health Sciences in KNUST, Kumasi; the capital city of Ashanti Region of Ghana. The school has its campus at Boadi, a suburb of Kumasi. The veterinary programme run by the School of Veterinary Medicine is a 6-years programme with students in first, second and third years of study classified as pre-clinical students whilst students in clinical years are those in fourth, fifth and sixth years of study. The Veterinary School used in this study, does not have a residence area on campus solely for veterinary students and in view of this, students bear the responsibility to search for accommodation which could be far or near to the campus.

## Study population

The study population in this current research comprised all the students of the School of Veterinary Medicine in Kwame Nkrumah University of Science and Technology. The overall population of the students in the School of Veterinary Medicine as at the time of data collection for this current study stood at a total of one hundred and eighty-five (185) students in different years of study. Characteristically, the student population comprises of different individuals with different socio-economic backgrounds and from different ethnicities including some international students from some African countries including Sierra Leone, Congo, Tanzania and Uganda.

## Sampling technique and sample size

The sampling technique employed for this study was the simple random sampling technique. This method of sampling technique was used so as to avoid biases on the part of the researcher as well as to ensure that each student was provided with the chance of being selected to be a part of the respondents for this study.

In determining the sample size, the Yamane formula [18] was used. The Yamane formular is stated as n = N/1+N (e)$^2$; where n is sample size. N is population size, e is sampling error constant, power of constant which is 0.05. For sample size calculation;

$$n = ?, \; N = 125, \; e = 0.05$$

Therefore,

$$n = 185/\left[1 + 185 \, (0.05)^2\right]$$
$$n = 125/\left[1 + \, 185(0.0025)\right]$$
$$n = 125/\left[1 + \, 0.4625\right]$$
$$n = 125/1.4625$$
$$n = 95$$

A total of 95 students were needed for the study based on the Yamane Formula. Nevertheless, the issue of COVID-19 caused a shutdown of the University and therefore questionnaires for the study were to be administered to students electronically. With this pandemic insight and due to the inability of all students to have internet connectivity at their various places of residence and localities, the total number of students who were able to have access to internet and were able to completely fill the questionnaire during the period of data collection served as the sample size for this study. This accounted for the differences with 74 students used in the study as compared to the actual ample size of 95 determined by the Yamane Formula above.

## Data collection instrument and procedure

In this study, a well-structured questionnaire was designed and designed with Google Forms served as the data collection instrument. The questionnaire comprised questions on the demographic characteristics of the respondents as well as certain practices such as their sleeping hours, exercise hours and study hours. The other section of the questionnaire comprised of questions on emotional exhaustion (EE), Depersonalisation (DP) and Reduced Personal Accomplishment (RPA). Questions on EE focused on the feeling of being emotionally and physically drained as a result of the workload of a veterinary student. Additionally, questions on DP focused on feelings of negativity and indifference towards others whilst questions on

RPA focused on the students' personal unfavourable judgement of themselves in terms of their academic performance.

The questionnaires were successfully administered to the respondents after obtaining their consent. The link to the questionnaires to be completed were delivered to the respondents electronically through electronic platforms (specifically WhatsApp). The responses to the questionnaires were retrieved from Google Forms in a Microsoft Excel format by the researcher after a one-month time frame (from June to July, 2020) has elapsed for students.

### Data analysis

Descriptive statistics were used to analyse the demographic characteristics (age, gender, year of study, marital status and residence) and practices of students as well as the responses of the students on emotional exhaustion, depersonalisation and reduced personal accomplishment in the form of frequencies, percentages, mean and standard deviations. The components of burnout were categorised into levels using the mean scores. For EE, low EE was a mean score of $\leq 17$ and moderate EE was a mean score of 18–29. For DP, low DP was a mean score of $\leq 6$, moderate DP was a mean score of 6–11 and high DP was a mean score of $\geq 12$. For RPA, low RPA was a mean score of $\leq 33$, moderate RPA was a mean score of 34–39 and high RPA was a mean score of $\geq 40$. The grading, scoring and categorization for each dimension of burnout ie. emotional exhaustion, depersonalization and reduced personal accomplishment was performed as described by some authors [1].

Pearson Chi-square test was used to determine the association between the demographic characteristics, practices of students and the level of burnout. With respect to the effect of demographic characteristics and practices of students on the burnout of students, multiple logistic regression analyses were used taking into considerations the odd ratios in order to ascertain the risk factors associated with the components of burnout. The outcome variable for the logistic regression model was a combined score on subscales of DP and RPA. The predictor variables included all the demographic characteristics (age, gender, level of study, marital status, residence status and practices of the students). Data analysis performed on all the data collected were performed using Statistical Package for Social Sciences (SPSS Version 20). All statistical analysis and significance were tested at 5% significant level or 95% confidence interval.

## Results

### Demographic characteristics of students

Results obtained for the demographic characteristics of the respondents from the data analysis are displayed in Table 1 below.

The findings indicated that out of the 74 respondents, 41 (55.4%) respondents were less than 22 years of age whilst the remaining 33 (44.6%) of the respondents were above 22 years of age. With respect to the sex distribution of the respondents, 54 (73%) of the respondents were males whilst the remaining 20 (27%) of the respondents were females. In terms of marital status of the respondents, 69 (94.6%) of the respondents were unmarried whilst the remaining 5 (5.4%) of the respondents were married. The level of study of the respondents was also considered in this study as shown in Table 1 below. The findings showed that 31 (41.9%) of the respondents were first year students, 7 (9.5%) of the respondents were second year students, 14 (18.9%) of the respondents were third year students, 10 (13.5%) were fourth year students, 9 (12.2%) of the respondents were fifth year students whilst the remaining 3 (4.1%) of the respondents were sixth year students. Results on the place of residence of veterinary students showed that 41.9% were living in rented apartments, 35.1% were living in dormitories, 12.2%

**Table 1. Results of demographic characteristics of respondents.**

| Variable | Category | Frequency (n) | Percentage (%) |
|---|---|---|---|
| Age of Students | <22 | 41 | 55.4 |
| | >22 | 33 | 44.6 |
| Gender of Students | Female | 20 | 27.0 |
| | Male | 54 | 73.0 |
| Marital Status of Students | Married | 5 | 5.4 |
| | Not Married | 69 | 94.6 |
| Year of Study | Preclinical (Year 1- Year 3) | 52 | 70.3 |
| | Clinical (Year 4-Year 6) | 22 | 29.7 |
| Residence of Students | Dormitory | 26 | 35.1 |
| | Hostel | 6 | 8.1 |
| | Rented apartment | 31 | 41.9 |
| | Traditional Hall | 2 | 2.7 |
| | With parent /family | 9 | 12.2 |

were living with their parents / families, 8.1% were living in hostels whilst 2.7% were living in traditional halls.

## Practices of students

Findings on the practices of veterinary students were considered in this study as shown in Table 2 below. Results on the weekly study hours showed that 50% studied between 25–40 hours, 31.1% studied less than 25 hours a week whilst 18.9% studied more than 40 hours in a week. It was observed that with respect to exercise hours amongst veterinary students, 64.9% spent less than 2 hours exercising, 28.4% spent between 2–5 hours exercising whilst 6.8% exercised above 5 hours. With sleep hours of students, majority of the students (62.2%) slept less than 6 hours whilst 37.8% slept between 6–10 hours. It was also indicated that 87.8% of the students were not working as they study whilst the remaining 12.2% were engaged in diverse forms of work alongside studying.

## Prevalence of burnout amongst students

Table 3 below summarizes the prevalence of burnout amongst veterinary students. It was observed that majority of the students (86.5%) students reported low levels of emotional exhaustion whilst 13.5% reported moderate levels of emotional exhaustion. The majority of

**Table 2. Results of practices of respondents.**

| Variable | Category | Frequency (n) | Percentage (%) |
|---|---|---|---|
| Weekly Study Hours | Less than 25 hours | 23 | 31.1 |
| | Between 25–40 hours | 37 | 50.0 |
| | Above 40 hours | 14 | 18.9 |
| Exercise Hours (weekly) | Less than 2 hours | 48 | 64.9 |
| | Between 2–5 hours | 21 | 28.4 |
| | Above 5 hours | 5 | 6.7 |
| Sleep Hours (daily) | Less than 6 hours | 46 | 62.2 |
| | Between 6–10 hours | 28 | 37.8 |
| Works whiles studying | Not Working | 65 | 87.8 |
| | Working | 9 | 12.2 |

**Table 3. Results on the prevalence of burnout amongst respondents.**

| Variable | Categories / Levels | Distribution (n (%)) | Mean subscales score |
|---|---|---|---|
| Emotional Exhaustion (EE) | Low (≤17) | 64 (86.5%) | 12.72 ± 4.46 |
| | Moderate (18–29) | 10 (13.5%) | |
| Depersonalization (DP) | Moderate (6–11) | 4 (5.4%) | 20.72 ± 6.51 |
| | High (≥12) | 70 (94.6) | |
| Reduced Personal Accomplishment (PA) | Low (≤33) | 57 (77.0%) | 27.96 ± 7.94 |
| | Moderate (34–39) | 12 (16.2%) | |
| | High (≥40) | 5 (6.8%) | |

the students (94.6%) reported a high level of depersonalization whilst 5.4% of the students reported a moderate level of depersonalization. Moreover, the majority of the students (77.0%) showed low levels of reduced personal accomplishment with 16.2% and 6.8% of the students showed moderate to high levels of reduced personal accomplishment respectively as shown in Table 3 below.

The mean scores for the various components of burnout indicated that on the average, the emotional exhaustion of the students was low with a mean score of 12.72 ± 4.46. Depersonalisation levels of the students in this study were high with a mean score of 20.72 ± 6.51. With reduced personal accomplishment, the mean score of 27.96 ± 7.94 indicated that the students were satisfied with their personal accomplishments as shown in Table 3 below.

## Risk factors of burnout

### Demographic characteristics and burnout

The prevalence of burnout amongst veterinary students in this study with respect to the demographic characteristics of the respondents were compared as summarised in Table 4 below. With respect to emotional exhaustion (EE), no student indicated high EE. For age, the majority of the students indicated low EE were less than 22 years old (38, 59.4%) as compared to 26 (40.6%) of the students who were older than 22 years. In addition, 7 (70%) of the students who indicated moderate EE were above 22 years of age. More females 6(60%) indicated moderate

**Table 4. Results on demographic characteristics and burnout levels.**

| Characteristic | Category | Emotional Exhaustion (EE) | | Depersonalisation (DP) | | Reduced Personal Accomplishment (RPA) | | |
|---|---|---|---|---|---|---|---|---|
| | | Low | Moderate | Moderate | High | Low | Moderate | High |
| Age (years) | <22 | 38 (59.4%) | 3 (30%) | 1 (25%) | 40 (57.1%) | 33(54.4%) | 4 (33.3%) | 4 (80%) |
| | >22 | 26 (40.6%) | 7 (70%) | 3 (75%) | 30 (42.9%) | 24(45.6%) | 8 (66.7%) | 1 (20%) |
| Sex | Male | 50 (78.1%) | 4 (40%) | 2 (50%) | 52 (74.3%) | 50(87.7%) | 3 (25%) | 2(41.6%) |
| | Female | 14 (21.9%) | 6 (60%) | 2 (50%) | 18 (25.7%) | 8(59.4%) | 9 (75%) | 3(59.4%) |
| Marital Status | Married | 4 (6.3%) | 1 (10%) | 3 (75%) | 2 (2.9%) | 4 (7%) | 0 (0%) | 0 (0%) |
| | Single | 60 (93.7%) | 9 (90%) | 1 (25%) | 68 (97.1%) | 53 (93%) | 12 (100%) | 5(100%) |
| Year of study | Preclinical | 45 (70.3%) | 7 (70%) | 3 (75%) | 49 (70%) | 38(66.7%) | 10 (83.3%) | 4 (80%) |
| | Clinical | 19 (29.7%) | 3 (30%) | 1 (25%) | 21 (30%) | 19(33.3%) | 2 (16.7%) | 1 (20%) |
| Residence | Dormitory | 25 (39.1%) | 1 (10%) | 2 (50%) | 24 (34.3%) | 23(40.3%) | 3 (25%) | 0 (0%) |
| | Parents /family | 7 (10.9%) | 2 (20%) | 0 (0%) | 9 (12.9%) | 3 (5.3%) | 2 (16.7%) | 4 (80%) |
| | Rented apartment | 26 (40.6%) | 5 (50%) | 0 (0%) | 31 (44.3%) | 25 (43.8%) | 5 (41.7%) | 1 (20%) |
| | Traditional Hall | 2 (3.1%) | 0 (0%) | 1 (25%) | 1 (1.4%) | 1 (1.8%) | 1 (8.3%) | 0 (0%) |
| | Hostel | 4 (6.3%) | 2 (20%) | 1 (25%) | 5 (7.1%) | 5 (8.8%) | 1 (8.3%) | 0 (0%) |

levels of EE than males 14 (40%). More students in the preclinical year (7, 70%) reported moderate levels of EE as compared to the clinical year students (3, 30%). More students in rented apartments (50%) reported moderate levels of EE as compared those living in dormitories (10%), traditional halls (0%), hostels (10%) and staying with their parents/ family (20%) Table 4.

Considering depersonalisation, more students who were less than 22 years (40, 57.1%) reported high levels of DP. It was observed that more male students (52, 74.3%) reported high DP as compared to 18 (25.7%) of the females. More students (49, 70%) in the preclinical year reported high levels of DP as compared to the clinical year students (21, 30%). It was further observed that more students in rented apartments (44.3%) reported high levels of EE as compared those living in dormitories (34.3%), traditional halls (1.4%), hostels (7.1%) and staying with their parents/ family (12.9%) (Table 4 above).

Results on reduced personal accomplishment showed that with respect to high RPA more students who were less than 22 years (80%) reported high levels of RPA as compared to the 20% of the students above 22 years of age. It was observed that female students (59.4%) reported high RPA as compared to the males (41.6%). More students (80%) in the preclinical year reported high levels of RPA as compared to the clinical year students (20%). It was also observed that more students living with their families or parents (80%) reported high levels of RPA as compared those living in dormitories (20%). However, with respect to moderate levels of RPA, more students living in rented apartments (41.7%) indicated moderate levels of RPA as compared to students in other types of residence (Table 4 above). The significance between the difference in the demographic characteristics on the components of burnout are presented in Table 5 below.

## Effects of demographic characteristics on burnout

Findings on the statistical significance of students' demographic characteristics on the high levels of DP, and RPA is displayed in Table 5 below. The findings revealed that there was no significant effect ($p > 0.05$) of age, sex, marital status on the high levels of depersonalisation and reduced personal accomplishments of the veterinary students in this study.

**Table 5. Tabulation of students' demographic characteristics and high levels of DP and RPA.**

| Characteristic | Category | DP | Chi-square value | P-value | RPA | Chi-square value | P-value |
|---|---|---|---|---|---|---|---|
| Age (years) | <22 | 40 (57.1%) | 1.542 | 0.190 | 4 (80%) | 1.558 | 0.204 |
| | >22 | 30 (42.9%) | | | 1 (20%) | | |
| Sex | Male | 52 (74.3%) | 2.519 | 0.321 | 2 (41.6%) | 1.329 | 0.240 |
| | Female | 18 (25.7%) | | | 3 (59.4%) | | |
| Marital Status | Married | 2 (2.9%) | 0.360 | 0.059 | 0 (0%) | 0.773 | 0.860 |
| | Single | 68 (97.1%) | | | 5 (100%) | | |
| Year of study | Preclinical | 49 (70%) | 0.448 | 0.034* | 4 (80%) | 0.390 | 0.000* |
| | Clinical | 21 (30%) | | | 1 (20%) | | |
| Residence | Dormitory | 24 (34.3%) | 0.173 | 0.015* | 0 (0%) | 0.180 | 0.028* |
| | Parents /family | 9 (12.9%) | | | 4 (80%) | | |
| | Rented apartment | 31 (44.3%) | | | 1 (20%) | | |
| | Traditional Hall | 1 (1.4%) | | | 0 (0%) | | |
| | Hostel | 5 (7.1%) | | | 0 (0%) | | |

* = p-value is significant at 5% significance level (95% Confidence interval).

The results also showed that there was found a significant effect (p<0.05) of the year of study of the veterinary students as well as the residence of the students on the high level of DP and RPA represented by significance values of p = 0.000 and p = 0.028 respectively. The findings show that the type of residence and year of study are significant risk factors of high levels of depersonalisation and reduced personal accomplishment among veterinary students.

## Prevalence of burnout with respect to practices of students

Results on prevalence of burnout amongst the veterinary students with respect to their practices have been presented in Table 6 below. No student indicated high level of emotional exhaustion.

For depersonalisation, the majority of the students (88.6%) who indicated high levels of DP were students who were not working while studying. More students (45.7%) who indicated high levels of DP studied between 25–40 hrs a week as compared to those who studied more than 40 hours a week (17.1%) and less than 25 hours a week (32.9%). Furthermore, more students who exercised less than 2 hours weekly (67.2%) indicated high levels of DP as compared to the other students who exercised more than 2 hours weekly (31.3%). In addition, more students who indicated high levels of DP slept less than 6 hours in a day (63.2%) as compared to those who slept between 6–10 hours in a day (32.8%) as shown in Table 5 above.

With reduced personal accomplishment (RPA), the majority of the students (88.6%) who indicated high levels of RPA were students who were not working whiles studying. More students (45.7%) who indicated high levels of RPA studied less than 25 hours a week as compared to those who studied more than 40 hours a week (0%) and between 25–40 hours a week (40%). Furthermore, more students who exercised less than 2 hours weekly (40%) and between 2–5 hours (40%) indicated high levels of RPA as compared to the other students who exercised more than 5 hours weekly (20%). In addition, more students who indicated high levels of RPA slept less than 6 hours in a day (60%) as compared to those who slept between 6–10 hours in a day (40%) as shown in Table 5 above.

## Effects of students' practices on burnout

The result on the effect of practices of students on high levels of DP and RPA is displayed in Table 7 below. The results revealed that there were no significant effects (p>0.05) of the working status, weekly study hours and exercise hours of the veterinary students on the high levels of depersonalisation and reduced personal accomplishments of the veterinary students in this

**Table 6. Results on student practices and burnout levels.**

| Characteristic | Category | Emotional Exhaustion | | Depersonalisation | | Reduced Personal Accomplishment | | |
|---|---|---|---|---|---|---|---|---|
| | | Low | Moderate | Moderate | High | Low | Moderate | High |
| Work status | Working | 7 (10.9%) | 2 (20%) | 1 (25%) | 8 (11.4%) | 7(9.7%) | 1(8.3%) | 1 (20%) |
| | Not working | 57 (89.1%) | 8 (80%) | 3 (75%) | 62 (88.6%) | 50 (69.4%) | 11(91.7%) | 4 (80%) |
| Weekly Study hours | < 25 hrs | 18 (28.1%) | 5 (50%) | 0 (0%) | 23(32.9%) | 16 (28.1%) | 4 (33.3%) | 3 (60%) |
| | 25–40 hrs | 33 (51.6%) | 4 (40%) | 2 (50%) | 32(45.7%) | 33 (57.9%) | 2 (16.7%) | 2 (40%) |
| | > 40 hrs | 13 (20.3%) | 1 (10%) | 2 (50%) | 12 (17.1%) | 8 (14%) | 6 (50%) | 0 (0%) |
| Exercise Hours (Weekly) | < 2 hrs | 42 (65.6%) | 6 (60%) | 1 (25%) | 47(67.2%) | 37 (64.9%) | 9 (75%) | 2 (40%) |
| | 2–5 hrs | 20 (31.3%) | 1 (10%) | 2 (50%) | 19 (27.1%) | 16 (28.1%) | 3 (25%) | 2 (40%) |
| | > 5 hrs | 2 (3.1%) | 3 (30%) | 1 (25%) | 4 (5.7%) | 4 (7%) | 0 (0%) | 1 (20%) |
| Sleep Hours (Daily) | < 6 hrs | 39 (60.9%) | 7 (70%) | 3 (75%) | 43 (61.4%) | 36 (63.2%) | 7 (58.3%) | 3 (60%) |
| | 6–10 hrs | 25 (39.1%) | 3 (30%) | 1 (25%) | 27(38.6%) | 21 (36.8%) | 5 (41.7%) | 2 (40%) |

**Table 7. Results on practices and burnout levels.**

| Characteristic | Category | DP | Chi-square value | P-value | RPA | Chi-square value | P-value |
|---|---|---|---|---|---|---|---|
| Work status | Working | 8 (11.4%) | 0.142 | 0.077 | 1 (20%) | 0.132 | 0.086 |
| | Not working | 62 (88.6%) | | | 4 (80%) | | |
| Weekly Study hours | < 25 hrs | 23(32.9%) | 3.341 | 0.104 | 3 (60%) | 0.456 | 0.234 |
| | 25–40 hrs | 32(45.7%) | | | 2 (40%) | | |
| | > 40 hrs | 12 (17.1%) | | | 0 (0%) | | |
| Exercise Hours (Weekly) | < 2 hrs | 47(67.2%) | 1.447 | 0.054 | 2 (40%) | 1.296 | 0.098 |
| | 2–5 hrs | 19 (27.1%) | | | 2 (40%) | | |
| | > 5 hrs | 4 (5.7%) | | | 1 (20%) | | |
| Sleep Hours | < 6 hrs | 43 (61.4%) | 1.863 | 0.035* | 3 (60%) | 0.176 | 0.049* |
| | 6–10 hrs | 27(38.6%) | | | 2 (40%) | | |

* = p-value is significant at 5% significance level (95% Confidence interval).

study. Though there were noticeable differences between the categories under each student practice risk factors for high DP and PA, these differences were not significant.

However, there was a statistically significant effect (p<0.05) of the sleep hours of students on the high levels of DP and RPA. This finding shows that the sleeping hours of students are significant risk factors of high levels of depersonalisation and reduced personal accomplishment among veterinary students.

Table 8 summarizes the regression analysis of burnout subscales (RPA and DP), demographics and practices of students. The year of study was a significant predictor of reduced personal achievement [OR = 4.28; 95% CI = 1.053–7.180; P-value 0.04*] as per multivariate analysis for demographic characteristics and reduced personal achievement. Similarly, year of study was a significant predictor of depersonalization [OR = 2.75.; 95% Cl 2.016–4.750; P-value 0.010*] as per multiple regression analysis for demographic characteristics and depersonalization.

The type of residence of students was a significant predictor of reduced personal achievement [OR = 0.29; 95% CI = 2.602–5.316; P-value 0.02*] as per multivariate analysis for

**Table 8. Logistic regression analysis for burnout subscales, students' demographics and student practices.**

| Parameter | RPA | | | DP | | |
|---|---|---|---|---|---|---|
| | OR | P-value | CI (95%) | OR | P-value | CI (95%) |
| **Demographic Characteristics** | | | | | | |
| Age | 1.34 | 0.65 | [0.753–5.423] | 0.13 | 0.58 | [0.439–2.742] |
| Sex | 2.52 | 0.12 | [0.519–2.003] | 1.09 | 0.25 | [0.216–2.043] |
| Marital Status | 0.65 | 0.89 | [0.433–1.696] | 0.18 | 0.62 | [1.643–3.218] |
| Year of study | 4.28 | 0.04* | [1.053–7.180] | 2.75 | 0.01* | [2.016–4.750] |
| Residence | 0.29 | 0.02* | [2.602–5.316] | 1.30 | 0.03* | [0.503–2.671] |
| **Practices of students** | | | | | | |
| Work Status | 1.69 | 0.56 | [1.117–3.430] | 6.43 | 0.06 | [0.605–4.078] |
| Study Hours | 1.73 | 0.07 | [0.950–2.406] | 2.09 | 0.66 | [0.785–5.320] |
| Exercise Hours | 0.89 | 0.09 | [1.352–4.007] | 2.2 | 0.41 | [2.653–10.541] |
| Sleep Hours | 4.32 | 0.014* | [4.306–9.001] | 1.24 | 0.000* | [1.349–6.301] |

OR- Odd Ratio, CI- Confidence interval,

* = p-value is significant at 5% significance level

demographic characteristics and reduced personal achievement. Similarly, residence of students was a significant predictor of depersonalization [OR = 1.30.; 95% Cl 0.503–2.671; P-value 0.03*] as per multiple regression analysis for demographic characteristics and depersonalization (Table 8).

Findings indicated that daily sleep hours of students were identified a significant predictor of reduced personal achievement [OR = 4.32; 95% CI = 4.306–9.001; P-value 0.014*] as per multivariate analysis for student practices and reduced personal achievement. Similarly, daily sleep hours of students was identified as a significant predictor of depersonalization [OR = 1.24.; 95% Cl 1.349–6.301; P-value 0.000*] as per multiple regression analysis for student practices and depersonalization (Table 8).

## Discussion

This current study investigated the prevalence and associated risk factors of burnout among veterinary students in Ghana. The outcome of the findings of this study indicated that on the average, EE of veterinary students was low as indicated by 64 (86.5%) of the students with mean score of 12.72 ± 4.46. RPA was also low as indicated by 57 (77.0%) of the students with a mean score of 27.96 ± 7.94 whilst DP levels of the veterinary students in this study were high as indicated by 70 (94.6%) of the students with a mean score of 20.72 ± 6.5. This finding shows that the level of burnout among veterinary students is not relatively high; however, the depersonalisation component of burnout is the most area experienced by the students. This finding is in sharp contrast to the findings of a study conducted by [10] who in his findings showed that there exists a high level of burnout among veterinary students in Australia. The differences in the prevalence of burnout could be as a result of differences in geographical location as well as the differences in the nature of how the veterinary medicine programme is executed in these two different countries. In Australian veterinary students, it is known that students spend a significant proportion of their time on extramural study (EMS) placements within veterinary clinics, working alongside veterinarians and their clients. Many students also have paid employment in veterinary practices, often as veterinary nurses and therefore more prone to burnout [10] as compared to veterinary students in Ghana who are not engaged in such rigorous activities or paid jobs.

This study also revealed that levels of emotional exhaustion experienced by the veterinary students were low to moderate with majority of the students experiencing low levels of emotional exhaustion. The relatively high depersonalisation among students in this study could be as a result of accumulated stress from relatively unsatisfied academic performances over time. This inadvertently could lead to the development of mostly negative consequences such as having panic disorders and facing severe anxiety or depression. Additionally, these students tend to develop feelings of negativity and indifference towards other students and even their lecturers. This finding in this study corroborates the findings of [16]. According to studies conducted by [16], veterinary medical students experience a lot of psychological and physiological changes during training especially those in their formative years. Academic stressors which include heavy workload, competitiveness with peers, unclear instructor expectations, and excessive worry about being less intelligent than peers are some of the triggers of depersonalization. In addition, many veterinary medical students experience stressful life events such as relational concerns, inadequate self-care regimens, long learning hours as well as physical health concerns which contribute to the unhealthy psychological changes which can result easily in depersonalization [19].

Findings on the effects of demographic characteristics of veterinary students on the high levels of DP and RPA showed that age did not have any significant effect on the high levels of

DP and RPA; though the students who were less the 22 years of age recorded high levels of DP (57.1%) and RPA (80%) as compared to the students who were older than 22 years of age (high DP = 42.9%; high RPA = 20%). This shows that burnout amongst veterinary students was not significantly affected by the age of the student though those in the lower age limits suffer the stress of depersonalisation and reduced accomplishments more.

The findings of this current study furthermore showed that there is no significant impact of gender on the prevalence of burnout among veterinary students although females more frequently reported a high level of RPA (59.4%) as compared to male students (41.6%) whilst the male students (74.3%) reported more frequently a high level of DP as compared to the females (25.7%). This finding corroborates the findings of Ilić Živojinović *et al.*, [9] who reported that though female students reported higher levels of burnout especially with emotional exhaustion, the effect of gender on burnout levels were not statistically significant. Notwithstanding, the findings of this study is in sharp contrast with the findings of Gelberg and Gelberg [20], who in their study on stress in veterinary students established that gender has a significant effect on stress and burnout levels with females being the most susceptible as compared to their male counterparts.

The findings of this study showed that the year of study of veterinary students had a significant impact on the level of burnout experienced by the students. This study revealed that the levels of high RPA (80%) and DP (70%) was more in the preclinical students as compared to the clinical students (RPA = 20%, DP = 30% respectively). This finding is consistent with the findings of Feras *et al.* [21], who in their report concerning burnout in medical students in Lebanon indicated that there is a high level of 75% burnout among preclinical medical students. This could be attributed to the fact that students in preclinical years are new to the various academic and non-academic stressors involved in being a student of veterinary school and hence are now adjusting to the pressures of balancing the rigorous academic course with extracurricular activities. Whereas students in the clinical years have been preconditioned and have adjusted to the academic stress involved in being a veterinary medical student.

The residence of students was found to have a significant effect on the prevalence of burnout in this study. Students who were residing in rented apartments (80%) as well as those living with their families (44.3%) recorded higher levels of DPA and RPA respectively as compared to those who lived in halls (1.4%) on the main KNUST campus. This finding could be explained by the mode of transportation of students from their places of residence to the Veterinary school campus which is located farther away from the main KNUST campus. Also, the financial constraint posed to most students in renting apartments, and hostels could explain this effect of place of residence on burnout levels. The findings of this study support the findings of Chigerwe *et al* [11] who indicated the type of living arrangements of students in California was associated with burnout.

In this study, the marital status of veterinary students did not have any significant effect (p>0.05) on the high levels of depersonalisation and reduced personal accomplishments of veterinary students. However, there were noticeable differences with high levels of DP and RPA recorded more in the unmarried students of 97.1% and 100% respectively. This difference could be as a result of the lack of emotional partners who can support them to cope in times where they feel unhappy about their academic performances as well as have negative thoughts about their programme of study.

Furthermore, this study revealed that some practices of students could serve as risk factors of burnout in their academic pursuit. The findings showed that daily sleep hours of students had significant effect on the prevalence of burnout. Veterinary students who slept less than 6 hours in a day in this study indicated high levels of DP (61.4%) and RPA (60%) as compared to those who slept between 6–10 hours in a day (high RPA = 40%; high DP = 38.6%). This

phenomenon could be attributed to the fact that sleep is significant to reenergising the body and the mind to deal with academic activities. Therefore, less amount of sleep will result in excessive tiredness of both the body and the mind which will affect academic performance as well as lead to low physical energy to carry-out daily activities effectively and efficiently.

There was also found no significant effects (p>0.05) of the working status, weekly study hours and exercise hours of the veterinary students on depersonalisation and reduced personal accomplishments.

## Conclusion

Low levels of burnout were observed among veterinary students in Ghana in this study in terms of low levels of emotional exhaustion and reduced personal achievement recorded in the study. Nevertheless, high levels of depersonalisation were observed among the students in this study. The year of study and the place of residence of veterinary students were found to be the associated demographic risk factors for burnout. Daily sleep hours of the veterinary students were also indicated as an associated risk factor of burnout among veterinary students in this study. It is recommended that veterinary students be given the necessary facilities and attention to deal with the increase in depersonalisation. In addition, based on the limitations of the study which involved lesser sample size and use of one of the two veterinary schools in Ghana, further studies should include all the veterinary schools in Ghana using larger sample size in order to provide a more detailed view on burnout in veterinary students.

## Author Contributions

**Conceptualization:** Benjamin Obukowho Emikpe.

**Data curation:** Ludwig Albert Nortey Botchway.

**Formal analysis:** Derrick Adu Asare, Richard Abeiku Bonney.

**Investigation:** Derrick Adu Asare.

**Methodology:** Richard Abeiku Bonney.

**Supervision:** Benjamin Obukowho Emikpe, Abigael Omowumi Emikpe.

**Writing – original draft:** Derrick Adu Asare.

**Writing – review & editing:** Benjamin Obukowho Emikpe, Abigael Omowumi Emikpe.

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
