## [Decision Letter · Decision Letter 0]

18 Jun 2021

PONE-D-20-27094

PREVALENCE AND ASSOCIATED RISK FACTORS OF BURNOUT AMONGST VETERINARY STUDENTS IN GHANA

PLOS ONE

Dear Dr. EMIKPE,

Thank you for submitting your manuscript to PLOS ONE. After careful consideration, we feel that it has merit but does not fully meet PLOS ONE’s publication criteria as it currently stands. Therefore, we invite you to submit a revised version of the manuscript that addresses the points raised during the review process.

The manuscript has been evaluated by two reviewers, and their comments are available below.

The reviewers have raised a number of major concerns. They feel the manuscript requires significant improvements to the English language and editing of this manuscript. They also suggest greater clarity in the methodological reporting of this study, and note that necessity for further detail on the methods and analyses, in addition to greater elaboration in the discussion section regarding the strengths, weaknesses and recommendations deduced from this study.

Could you please carefully revise the manuscript to address all comments raised?

We look forward to receiving your revised manuscript.

Kind regards,

Avanti Dey, PhD

Staff Editor

PLOS ONE

Journal Requirements:

Additional Editor Comments (if provided):

Reviewers' comments:

Reviewer's Responses to Questions

**Comments to the Author**

1. Is the manuscript technically sound, and do the data support the conclusions?

Reviewer #1: Partly

Reviewer #2: No

2. Has the statistical analysis been performed appropriately and rigorously? 

Reviewer #1: Yes

Reviewer #2: No

3. Have the authors made all data underlying the findings in their manuscript fully available?

Reviewer #1: Yes

Reviewer #2: No

4. Is the manuscript presented in an intelligible fashion and written in standard English?

Reviewer #1: Yes

Reviewer #2: No

5. Review Comments to the Author

Reviewer #1: English editing

Table 2 indicate exercise hours if per day or week as well as sleeping hours

Works “while” and not whiles

Demographic characteristics and burnout 7 (70%) as well as the rest of the percentages in the same paragraph.

Discussion and not discussion

use abbreviations after being defined

In the conclusion, burnout was noted due to high depersonalization so remove low levels of exhaustion and personal achievement.

I highly recommend rewriting the conclusion as it doesn’t reflect the strengths, weaknesses and recommendations deduced from this study.

Reviewer #2: The manuscript under review explored the prevalence and associated risk factors among veterinary students in Ghana. Although the researchers tackle an important area of study (burnout in veterinary students), there are some gaps in logic, omissions, and underdeveloped sections in the paper that I believe warrant consideration by the authors and editor. These more substantial edits/suggestions are provided in the list below and in the attached hard copy version. At a more basic level, there are typographical and grammatical errors throughout the document which detract from the clarity of content. I have attached my hard copy edits, which include suggestions for fixing these more minor errors.

1. In the Method section (p. 5), I am wondering how the researchers resolved the issue of the discrepancy between what was needed for Yamane formula and what was actually obtained? This limitation might be worth discussing in more detail in the discussion section (along with any other limitations of the study).

2. On pages 12 – 18, there are very limited descriptions of the statistics tests that were conducted, and the results of these tests are not well described. To remedy, I would suggest describing which tests were conducted at the outset of a paragraph or section (e.g., chi-square or logistic regression) and then include more statistical details for each of the tests (e.g., for a logistic regression you would report the overall model statistics and then individual coefficients, p values, and odds ratios).

3. In the discussion and/or the introduction section, can you provide specific percentages of the average rates of EE, RFA, and DP for the reader to make comparisons between the sample in this study and the larger population.

4. The authors state on p. 18 that “burnout is expressed the most due to depersonalization.” This doesn’t seem entirely accurate given that two of the Burnout components were actually lower than.

5. On p. 19, the discussion about why depersonalization is higher in this sample seems underdeveloped. I would like to see the authors explore the possible reasons for this manifestation of Burnout (and not the others) in the context of this population and the stressors and life experiences they have.

6. The finding that sleep may result in tiredness and a lack of productivity doesn’t seems quite simple and not particularly novel. I’m wondering if the authors can expand on the implications for this connection as it relates to students and practicing vets.

7. The statement that “burnout was observed” at the outset of the conclusion section seems somewhat misleading because you only found that one component of Burnout was observed and in fact the other two components were actually higher than average. I would suggest that this is rephrased here to capture the nuances of the results and, again, would suggest expanding on how and why depersonalization appears to the form that burnout takes with this group.

6. PLOS authors have the option to publish the peer review history of their article (what does this mean?). If published, this will include your full peer review and any attached files.

Reviewer #1: No

Reviewer #2: No

---

## [Author Response · Author response to Decision Letter 0]

10 Aug 2021

31st July, 2021

Manuscript PONE-D-20-27094

Staff Editor

PLOS ONE

Dear Avanti Dey (PhD),

Thank you for giving us the opportunity to submit a revised draft of the manuscript “PREVALENCE AND ASSOCIATED RISK FACTORS OF BURNOUT AMONGST VETERINARY STUDENTS IN GHANA” for publication in PLOS ONE. We appreciate the time and effort that you and the reviewers dedicated to providing feedback on our manuscript and are grateful for the insightful comments on and valuable improvements to our paper. We have incorporated most of the suggestions made by the reviewers. Those changes are highlighted within the manuscript. Please see below, in blue, for a point-by-point response to the editor’s and reviewers’ comments and concerns. 

Reviewers' Comments to the Authors:

Reviewer 1

1. Table 2 indicate exercise hours if per day or week as well as sleeping hours

Works “while” and not whiles

Demographic characteristics and burnout 7 (70%) as well as the rest of the percentages in the same paragraph.

Discussion and not discussion

use abbreviations after being defined

Authors’ response: The reviewer is correct, and we have made the necessary grammatical error corrections as indicated. The frequency of exercise and sleeping hours have been specified in our revised manuscript. 

2. In the conclusion, burnout was noted due to high depersonalization so remove low levels of exhaustion and personal achievement. 

I highly recommend rewriting the conclusion as it doesn’t reflect the strengths, weaknesses and recommendations deduced from this study.

Authors’ response: The recommendation given has been considered and the authors have made changes to the conclusion to reflect the strengths, weaknesses and recommendations as indicated by the reviewer. 

Reviewer 2

1. In the Method section (p. 5), I am wondering how the researchers resolved the issue of the discrepancy between what was needed for Yamane formula and what was actually obtained? This limitation might be worth discussing in more detail in the discussion section (along with any other limitations of the study).

Authors’ response: The cause for the discrepancy observed in the actual sample size used and that obtained from the Yamane Formula have been explained in detail in the Method section under the heading “Sample size”. 

2. On pages 12 – 18, there are very limited descriptions of the statistics tests that were conducted, and the results of these tests are not well described. To remedy, I would suggest describing which tests were conducted at the outset of a paragraph or section (e.g., chi-square or logistic regression) and then include more statistical details for each of the tests (e.g., for a logistic regression you would report the overall model statistics and then individual coefficients, p values, and odds ratios).

Authors’ response: The detailed descriptions of the statistics tests have been previously been spelt out in the Method section under the subsection “Data Analysis” where all tests have been spelt out and what they were used for. In addition, the results on the logistic regression and chi-square tests have been incorporated in the results section as recommended by the reviewer. 

3. In the discussion and/or the introduction section, can you provide specific percentages of the average rates of EE, RFA, and DP for the reader to make comparisons between the sample in this study and the larger population.

Authors’ response: The specific percentages of the average rates of EE, RFA and DP have been included in the revised manuscript. 

4. The authors state on p. 18 that “burnout is expressed the most due to depersonalization.” This doesn’t seem entirely accurate given that two of the Burnout components were actually lower than.

Authors’ response: The authors have revised this expression and have rectified and rephrased the statement to reflect the true findings of this study. 

5. On p. 19, the discussion about why depersonalization is higher in this sample seems underdeveloped. I would like to see the authors explore the possible reasons for this manifestation of Burnout (and not the others) in the context of this population and the stressors and life experiences they have.

Authors’ response: The recommendation given has been considered and the authors have developed the said deficit in writing that you have stated. More reasons have been adduced to why depersonalization was higher in this study. 

6. The finding that sleep may result in tiredness and a lack of productivity doesn’t seems quite simple and not particularly novel. I’m wondering if the authors can expand on the implications for this connection as it relates to students and practicing vets

Authors’ response: The connection between sleep and tiredness might not be novel but in our study, however, it adds on to the substantive argument of some the source of stresses that could have easily contribute to burnout in veterinary students. This study sought to explore the risk factors associated with burnout and so on a broader view the authors thought of taking a look at how long students are able to get enough rest as a suggestive predictor of burnout. 

7. The statement that “burnout was observed” at the outset of the conclusion section seems somewhat misleading because you only found that one component of Burnout was observed and in fact the other two components were actually higher than average. I would suggest that this is rephrased here to capture the nuances of the results and, again, would suggest expanding on how and why depersonalization appears to the form that burnout takes with this group.

Authors’ response: The recommendation given has been considered and the authors have rephrased the conclusion to reflect the findings of this study and not to mislead.

---

## [Decision Letter · Decision Letter 1]

24 Jun 2022

PONE-D-20-27094R1PREVALENCE AND ASSOCIATED RISK FACTORS OF BURNOUT AMONGST VETERINARY STUDENTS IN GHANAPLOS ONE

Dear Dr. EMIKPE,

Thank you for submitting your manuscript to PLOS ONE. After careful consideration, we feel that it has merit but does not fully meet PLOS ONE’s publication criteria as it currently stands. Therefore, we invite you to submit a revised version of the manuscript that addresses the points raised during the review process.

The manuscript under review explored the prevalence and associated risk factors among veterinary students in Ghana. Although the researchers tackle an important area of study (burnout in veterinary students), there are some gaps in logic, omissions, and underdeveloped sections in the paper that I believe warrant consideration by the authors and editor. These more substantial edits/suggestions are provided in the list below and in the attached hard copy version. At a more basic level, there are typographical and grammatical errors throughout the document which detract from the clarity of content. I have attached my hard copy edits, which include suggestions for fixing these more minor errors.

1. In the Method section (p. 5), I am wondering how the researchers resolved the issue of the discrepancy between what was needed for Yamane formula and what was actually obtained? This limitation might be worth discussing in more detail in the discussion section (along with any other limitations of the study).

2. On pages 12 – 18, there are very limited descriptions of the statistics tests that were conducted, and the results of these tests are not well described. To remedy, I would suggest describing which tests were conducted at the outset of a paragraph or section (e.g., chi-square or logistic regression) and then include more statistical details for each of the tests (e.g., for a logistic regression you would report the overall model statistics and then individual coefficients, p values, and odds ratios).

3. In the discussion and/or the introduction section, can you provide specific percentages of the average rates of EE, RFA, and DP for the reader to make comparisons between the sample in this study and the larger population.

4. The authors state on p. 18 that “burnout is expressed the most due to depersonalization.” This doesn’t seem entirely accurate given that two of the Burnout components were actually lower than.

5. On p. 19, the discussion about why depersonalization is higher in this sample seems underdeveloped. I would like to see the authors explore the possible reasons for this manifestation of Burnout (and not the others) in the context of this population and the stressors and life experiences they have.

6. The finding that sleep may result in tiredness and a lack of productivity doesn’t seems quite simple and not particularly novel. I’m wondering if the authors can expand on the implications for this connection as it relates to students and practicing vets.

7. The statement that “burnout was observed” at the outset of the conclusion section seems somewhat misleading because you only found that one component of Burnout was observed and in fact the other two components were actually higher than average. I would suggest that this is rephrased here to capture the nuances of the results and, again, would suggest expanding on how and why depersonalization appears to the form that burnout takes with this group.

We look forward to receiving your revised manuscript.

Regards,

Mohammad Hossein Ebrahimi

PLOS ONE

Journal Requirements:

Additional Editor Comments (if provided):

Reviewers' comments:

Reviewer's Responses to Questions

**Comments to the Author**

1. If the authors have adequately addressed your comments raised in a previous round of review and you feel that this manuscript is now acceptable for publication, you may indicate that here to bypass the “Comments to the Author” section, enter your conflict of interest statement in the “Confidential to Editor” section, and submit your "Accept" recommendation.

Reviewer #1: All comments have been addressed

2. Is the manuscript technically sound, and do the data support the conclusions?

Reviewer #1: Yes

3. Has the statistical analysis been performed appropriately and rigorously? 

Reviewer #1: Yes

4. Have the authors made all data underlying the findings in their manuscript fully available?

Reviewer #1: Yes

5. Is the manuscript presented in an intelligible fashion and written in standard English?

Reviewer #1: Yes

6. Review Comments to the Author

Reviewer #1: (No Response)

7. PLOS authors have the option to publish the peer review history of their article (what does this mean?). If published, this will include your full peer review and any attached files.

Reviewer #1: No

---

## [Author Response · Author response to Decision Letter 1]

28 Jun 2022

RESPONSE TO REVIEWER’S COMMENTS ON PREVALENCE AND ASSOCIATED RISK FACTORS OF BURNOUT AMONGST VETERINARY STUDENTS IN GHANA

Comment 1: In the Method section (p. 5), I am wondering how the researchers resolved the issue of the discrepancy between what was needed for Yamane formula and what was actually obtained? This limitation might be worth discussing in more detail in the discussion section (along with any other limitations of the study).

Response: The discrepancy between the sample size used and the actual sample size obtained from the Yamane formula has been explained in the methodology aspect under the sample size section in the reviewed manuscript. 

Comment 2: On pages 12 – 18, there are very limited descriptions of the statistics tests that were conducted, and the results of these tests are not well described. To remedy, I would suggest describing which tests were conducted at the outset of a paragraph or section (e.g., chi-square or logistic regression) and then include more statistical details for each of the tests (e.g., for a logistic regression you would report the overall model statistics and then individual coefficients, p values, and odds ratios).

Response: This suggestion by the reviewer has been considered by the authors. The statistical tests conducted have been described in details in the data analysis section in the reviewed manuscript. 

Comment 3: In the discussion and/or the introduction section, can you provide specific percentages of the average rates of EE, RFA, and DP for the reader to make comparisons between the sample in this study and the larger population.

Response: The specific percentages of the average rates of EE, RFA, and DP have been provided appropriately to enhance comparison between the sample in this study. This has been added in the new manuscript under the discussion section. 

Comment 4: The authors state on p. 18 that “burnout is expressed the most due to depersonalization.” This doesn’t seem entirely accurate given that two of the Burnout components were actually lower than.

Response: The statement “burnout is expressed the most due to depersonalization” has been looked at. The write-up in that section has been re-written to reflect the findings on the burnout components in this study. This has been highlighted in the revised manuscript with tracked changes. 

Comment 5: On p. 19, the discussion about why depersonalization is higher in this sample seems underdeveloped. I would like to see the authors explore the possible reasons for this manifestation of Burnout (and not the others) in the context of this population and the stressors and life experiences they have.

Response: The authors have acknowledged the concern of the reviewer. In view of this, authors have provided possible reasons for depersonalization being the manifestation of burnout amongst the students in this study. This adjustment is found in the discussion section in the revised manuscript with tracked changes. 

Comment 6: The finding that sleep may result in tiredness and a lack of productivity doesn’t seems quite simple and not particularly novel. I’m wondering if the authors can expand on the implications for this connection as it relates to students and practicing vets.

Response: The authors have provided further explanations on sleep affecting productivity and the academic performance of students and practicing veterinarians but rather focusing more on veterinary students. 

Comment 7: The statement that “burnout was observed” at the outset of the conclusion section seems somewhat misleading because you only found that one component of Burnout was observed and in fact the other two components were actually higher than average. I would suggest that this is rephrased here to capture the nuances of the results and, again, would suggest expanding on how and why depersonalization appears to the form that burnout takes with this group.

Response: The conclusion has been recaptured and in the revised manuscript to capture the nuances of the results and have slightly expanded on how and why depersonalization appears to the form that burnout takes in this study.

---

## [Editor Report · Decision Letter 2]

1 Jul 2022

PREVALENCE AND ASSOCIATED RISK FACTORS OF BURNOUT AMONGST VETERINARY STUDENTS IN GHANA

PONE-D-20-27094R2

Dear Dr. EMIKPE,

We’re pleased to inform you that your manuscript has been judged scientifically suitable for publication and will be formally accepted for publication once it meets all outstanding technical requirements.

Kind regards,

Mohammad Hossein Ebrahimi

Academic Editor

PLOS ONE
---

## [Editor Report · Acceptance letter]

7 Jul 2022

PONE-D-20-27094R2 

Prevalence and associated risk factors of burnout amongst veterinary students in Ghana 

Dear Dr. Emikpe:

I'm pleased to inform you that your manuscript has been deemed suitable for publication in PLOS ONE. Congratulations! Your manuscript is now with our production department. 

Kind regards, 

on behalf of

Dr. Mohammad Hossein Ebrahimi 

Academic Editor

PLOS ONE